# Odontogenic-Related Head and Neck Infections: From Abscess to Mediastinitis: Our Experience, Limits, and Perspectives—A 5-Year Survey

**DOI:** 10.3390/ijerph20043469

**Published:** 2023-02-16

**Authors:** Resi Pucci, Andrea Cassoni, Daniele Di Carlo, Piero Bartolucci, Marco Della Monaca, Giorgio Barbera, Michele Di Cosola, Antonella Polimeni, Valentino Valentini

**Affiliations:** 1Department of Oral and Maxillofacial Sciences, Sapienza University of Rome, Via Caserta 6, 00161 Rome, Italy; 2Oncological and Reconstructive Maxillo-Facial Surgery Unit, Policlinico Umberto I, Viale del Policlinico 155, 00161 Rome, Italy; 3Surgical Sciences and Emergency Department, AOU Policlinico Umberto I Sapienza University of Rome, Viale del Policlinico 155, 00161 Rome, Italy; 4Department of Clinical and Experimental Medicine, Università Degli Studi di Foggia, 71122 Foggia, Italy

**Keywords:** head and neck infections, odontogenic infections, deep neck infections, mediastinitis, submandibular abscesses, dental abscess

## Abstract

Background: Head and neck infections are commonly caused by affections with an odontogenic origin. Untreated or non-responsive to treatment odontogenic infections can cause severe consequences such as localized abscesses, deep neck infections (DNI), and mediastinitis, conditions where emergency procedures such as tracheostomy or cervicotomy could be needed. Methods: An epidemiological retrospective observational study was performed, and the objective of the investigation was to present a single-center 5-years retrospective analysis of all patients admitted to the emergency department of the hospital Policlinico Umberto I “Sapienza” with a diagnosis of odontogenic related head and neck infection, observing the epidemiological patterns, the management and the type of surgical procedure adopted to treat the affections. Results: Over a 5-year period, 376,940 patients entered the emergency room of Policlinico Umberto I, “Sapienza” University of Rome, for a total of 63,632 hospitalizations. A total of 6607 patients were registered with a diagnosis of odontogenic abscess (10.38%), 151 of the patients were hospitalized, 116 of them were surgically treated (76.8%), and 6 of them (3.9%) manifested critical conditions such as sepsis and mediastinitis. Conclusions: Even today, despite the improvement of dental health education, dental affections can certainly lead to acute conditions, necessitating immediate surgical intervention.

## 1. Introduction

Head and neck infections are often linked to an odontogenic origin. Odontogenic infections represent the primary source of deep neck infections (38.8–49%) [1] and 89% of cases of severe multi-space infections [2]. Dental infections, if not treated, can lead to potential severe consequences that need to be managed with an interdisciplinary treatment, including intensive care unit, thoracic surgery, and ENT. These remarkable conditions, arising from local extension of dental infections, can have significant morbidity and potential mortality [3,4]. When odontogenic infections are no longer limited to the oral cavity, they can cause notable locoregional complications, increasing the severity of the pathological status, such as submandibular abscesses, which could evolve into a phlegmon or mediastinitis [4].

Although oral health is widely discussed and remains a very important social issue, serious complications from oral infections do not only involve developing countries, as might be expected [3]. The factors connected to the progression of odontogenic infections are on one hand patient-related, such as poor oral hygiene, metabolic status, and self-medication and, on the other hand, all the aspects related to the public health, such as the primary care, the lack of prevention, and the inadequate antibiotic therapies [5]. The delay of the treatment is another factor that should be considered, predisposing a patient affected by an odontogenic infection to a potential hospitalization or to the onset of serious complications [6,7]. Regardless of the region of the infection or the predisposing factors, it is noticeable that complicated odontogenic infections represent a frequent cause of access to the emergency department, and subsequently, a portion of these patients need to be hospitalized and subjected to more or less invasive medical and surgical treatments, such as dental extraction, extraoral drainage, cervicotomy, and mediastinum drainage. Lately, increasing rates of ER visits, as well as hospitalization of patients affected by odontogenic related infections, have been evaluated by public health facilities in the United States and Europe [8,9,10] and they are also becoming a consistent economic concern [11]. Odontogenic infections can range from localized infections that require minimal treatment, to severe life-threatening infections [12], and are also caused by a spreading of the purulent inflammation through communicating regions in the head and neck, with a high risk of sensitive anatomical region involvement, such as the brain [13] and the orbit, and have potential lethal outcomes [14,15]. According to Opitz et al., 4% of the patients affected by odontogenic infections receive surgical treatment and in 1.7% of the cases intensive postoperative medical therapy is required, with a documented lethal outcome of 0.12% [4]. According to Blankson et al., the incidence of odontogenic related infections among patients attending the dental clinic of the hospital is 40.3%, with the percentage of patients affected by dentoalveolar abscesses being 6.2%, with Ludwig’s angina representing the major kind of presentation of the spreading of the infection (52%); furthermore, a fatality rate of 5.8% has been described [16]. According to previous literature, males are the most affected, with a predominance ranging from mild [17] to significant [18,19]. However, these infections also have a high risk of sepsis and fatal complications for pregnant women: maternal mortality reaches 5.8% and that of the fetus to 13% [20]. Furthermore, these kinds of infections are the second leading cause of acute mediastinitis (AM), with the mortality rate remaining at the level of 18–44% over the past three decades [21]

These affections, their complications, and also their treatment, represent a public health concern [19] and a significant problem for the national health system, with up to 9.2% of patients needing urgent dental care in Hamburg, Germany [3]. Thomas et al. stated that admissions in the emergency department for a dental abscess drainage almost doubled over a 10-year period (1998–2006) and increased by 60% between 2005 and 2014, in Bristol, UK [22]. The absence of certain data on the number of patients who access the emergency room (ER) due to dental infections and how many of these ones be managed in the dental clinic does not allow for a significant assessment of the problem. The lack of guidelines and risk scales to stratify patients does not allow for identification of non-critical patients, who can be referred to outpatient facilities without saturating emergency units.

The aim of this investigation is to analyze the amount of accesses in the emergency room for odontogenic infections in five years, obtaining the epidemiological characteristics and tracing the path of the patients to calculate the percentage of serious complications and the mortality rate. The hospital Policlinico Umberto I had a number of 140,000 accesses per year to the Emergency and Acceptance Department before the COVID-19 emergency and remains the largest hospital in Europe at present.

## 2. Materials and Methods

An epidemiological retrospective observational study was conducted from the 1 January 2015 to the 31 December 2019. The analysis was performed for a five-year period, concerning all of the patients who accessed the emergency room of Policlinico Umberto I, Rome with a diagnosis of odontogenic infection. The number of patients that have been discharged from the hospital with oral antibiotic therapy, or sent to medical or dental clinic for treatment, or the number of patients that have been hospitalized and surgically treated, has been documented. Demographic characteristics, origin, primary dental care, and hospitalization data have been observed. Epidemiological data, surgical procedures used, and antibiotic therapy have been collected. The study was conducted in accordance with the Declaration of Helsinki and approved by the Institutional Review Board of Department of Oral and Maxillofacial Sciences, Sapienza University of Rome (N. 18/2022 Prot. no. 0000210 of 07/02/2022). The data collection system of the Emergency Department of Policlinico Umberto I, GIPSE, was used to identify the patients, including all patients with an acceptance diagnosis of “periapical abscess without sinus involvement, periapical abscess with sinus involvement, sinusitis, neck abscess, submandibular abscess, deep neck infection, other phlegmons and face abscesses, other phlegmons and abscesses of the neck, other diseases of the pharynx and nose, mediastinitis” using the relative associated code of the International Classification of Diseases (ICD). Once the total number of acceptance diagnoses has been identified, it was confirmed by checking the medical discharge diagnosis code. In this way, all patients hospitalized following an infection of odontogenic origin have been included. Epidemiological characteristics, age, sex, complications, number of patients treated, type of surgical procedure, and number of surgical treatments were collected. Pediatric patients were excluded from the study as they referred to the pediatric emergency department. All data were finally recorded for all those subjects affected by severe odontogenic infections admitted in the Maxillofacial Surgery Unit and surgically treated. Statistical analysis: the variables have been evaluated with the SPSS Software, version 26.0 and Fisher exact test and *t*-test were adopted. Statistical significance has been set at *p* < 0.05.

## 3. Results

Over a 5-year period, 376,940 patients had the first evaluation in the ER of Policlinico Umberto I, “Sapienza” University of Rome; 63,632 of them were hospitalized. Pediatric patients have been excluded from the study. A total of 6607 patients (10.38% of the of accesses) out of a total of 63,635 were admitted to the emergency department with a diagnosis of odontogenic infection, and for 151 of them hospitalization was confirmed. A total of 116 patients received surgical treatment. A total of 6607 patients entered the emergency room affected by an odontogenic related infection between 2015 and 2019: 36 patients during 2015, 620 patients during 2016, 2232 patients during 2017, 1880 patients during 2018, and 1839 patients during 2019. The definitive diagnosis of the patients who accessed the emergency department with an odontogenic infection had the following distribution frequencies: 0.2% of the patients were diagnosed with other diseases of the pharynx and nasopharynx (with a total of 13 patients affected), 1.5% with other phlegmons and abscesses of the neck (with a total of 114 patients affected), 2% with other phlegmons and abscesses of the head (with a total of 134 patients affected), 7.4% with periapical abscess with sinus involvement (with a total of 491 patients affected), 88.7% with periapical abscess without sinus involvement (with a total of 5863 patients affected), and 0.1% of the patients were diagnosed with odontogenic related mediastinitis (a total of 5 patients). Demographic characteristics of the patients such as sex and age have been reported in Table 1, with a total of 3010 female patients (45.6%) and 3597 males (54.4%). The mean age was 40.6 years old (SD ± 15.5).

The nationality of the patients can be observed in Figure 1, with a majority of Italian patients (4545 patients) scoring 68.8% of the sample, followed by Romanian patients, being 8.8% of the subjects (579 patients), while 2% of the patients were Bengalese (133 patients), and a smaller percentage of patients from many other countries was present: 1.8% from Ukraine with 120 patients, 1.6% from Egypt with 106 patients, 1.2% from Peru with 75 patients, 1.1% from Moldova with 72 patients affected, and finally, 1% from Morocco with 64 patients. Triage tags were also observed, as shown in Figure 2, with a total of 7 patients (0.1%) with critical conditions (red tags), 263 patients (4%) with moderately critical conditions (yellow tags), 4281 patients with green tags (64.8%), and 2056 white tags (31.1%).

A total of 5871 patients were discharged from the hospital and sent home (88.8%). A total of 517 patients (7.8%) were discharged from the hospital with oral antibiotic therapy and sent to a medical or dental clinic for treatment, followed by definitive conservative dental treatment (DCDT), 151 patients were hospitalized (2.3%) for intravenous antibiotic therapy with or without surgical treatment.

The mean age of hospitalized patients was 39.9 years old (SD ± 15.8), with a total of 55 female patients (36.42%) and 96 males (63.57%). Mean hospitalization time was 8.4 days (SD ± 5.4). Patients had diagnosis of abscess in 53.7% of cases (81 patients), deep neck infection (DNI) in 26.5%, (40 patients), Ludwig’s angina 16.5% (25 patients), and mediastinitis in 3.3% (5 patients). The data have been reported in Table 2.

One hundred and sixteen patients were surgically treated (76.8%). In the group of patients surgically treated the mean age was 37.4 years old (SD ±16.8); 73 were male (63%) and 43 female (37%). Extraoral drainage was performed in 96 patients (83%), dental extraction in 103 patients (89%), and in 15 patients cervicotomy was necessary (13%). Five patients presented with mediastinitis, and all were surgically treated by cervicotomy and mediastinum drainage together with thoracic surgeons; three of these patients underwent more than one surgical procedure (1.98% of the hospitalized patients), drains wereplaced, and all the compartments were explored and rinsed with lavages. In the five cases of mediastinitis, neck and superior mediastinum drainage was performed from a cervical approach, in three patients the infection descended at the level of tracheal carina, and the mediastinum was drained by performing a thoracotomy. Two patients with odontogenic mediastinitis died (1.3% of the hospitalized patients). Tracheostomy was performed in three patients in the emergency room (1.98% of the hospitalized patients). Statistical analysis: the variables were analyzed using the SPSS Software and Fisher exact test and *t*-test were adopted.

## 4. Discussion

Odontogenic infections represent a common oral and maxillofacial disease, they are a public health concern and may have potential life-threatening complications [22]. Treating these conditions is mandatory to avoid serious complications that could be lethal due to compromised respiratory space and mediastinum or blood contamination which can lead to sepsis [23]. The first objective of this investigation was to identify the amount of ER accesses during the year due to odontogenic infections. A total of 10.38% of the patients who accessed the emergency unit in this 5-years retrospective study had an infection of odontogenic origin. This high value agrees with what is described by Cachovan, who in seven years collected a sample of 5357 patients out of 58 161 who accessed the emergency unit with an odontogenic infection in Germany, 9.2% of the sample [3]. Odontogenic infections represent the cause of access to emergency units in 10% of cases, but only 2.28% of those patients required hospitalization;an important value for both the serious complications they can bring and for the public expense they represent [3]. Thomas et al. stated that admissions in the emergency department for a dental abscess drainage almost doubled over a 10-year period (1998–2006) and increased by 60% between 2005 and 2014 [22], costing the NHS around £275/day [23,24,25].

The investigation included 6607 patients, 54,4% of them were male, and 45.6% were female, with a mean age of 40.6 (SD ± 15.5). The largest sample was represented by males, with 54.4%, in agreement with Cachovan et al.; one possible reason for this observation could be that men appear to have worse oral health than women, and to have less attention to preventative health care visits [3]. A total of 68.8% of the patients of the sample were Italians, 8.8% of them were Romanians, and patients from other countries were present in smaller percentages (Bangladesh 2%, Egypt 1.6%, Morocco 1%, Moldova 1.1%, Peru 1.1%, Ukraine 1.8%). These data reflect the heterogeneity of the Italian population, it is also important to investigate the socioeconomic status of these patients. Indeed, as argued by Wang et al. these infections are more common in underserved patients without access to healthcare, patients who often have access to healthcare through the emergency department of publicly funded hospitals [19].

Most of these patients show affections that could be easily managed in the outpatient setting, while other conditions can potentially quickly become life-threatening complications [22]. A total of 5871 patients were discharged from the hospital and sent home. For 517 of 6607 patients in the study (7.8%), the hospitalization was not necessary, but they underwent dental evaluation, antibiotics treatment, and they referred to the dental clinic for a definitive conservative dental treatment (DCDT), in agreement with Cachovan et al. who directed 69.1% of the sample to non-surgical treatment [3]. This figure highlights how only a portion of the patients who access the emergency unit actually require hospital management. As argued by many Authors, the presence of dentists’ service of emergency management of odontogenic infections and effective protocols or guidelines would considerably reduce the load in public hospitals [3]. In fact, guidelines on the management of oral cavity infections could help dentists in the management of these patients in the dental clinic by reducing the influx into the ER. A total of 88.8% of these patients could be managed in an outpatient setting and with effective risk scales they could be referred directly to dental clinics to continue treatment. For example, the “Red flags” (as signs of systemic illness, voice changes, elevated tongue or floor of mouth, dyspnea, dysphagia, trismus) suggested in the study by Roy et al., who proposed a telephone questionnaire for the first evaluation of the patient by the maxillofacial surgeon to identify the severity of the symptoms [24]. This questionnaire, proposed in the UK, should be validated and standardized so that it can be applied widely. A total of 9% of the total access reduction in the ER would certainly also lead to a lower expenditure of economic resources by the NHS, but further studies are needed to demonstrate this hypothesis.

A total of 151 patients needed to be hospitalized, 116 underwent surgical treatments (76.8%), and 6 of them (3.9%) showed critical consequences such as sepsis and mediastinitis. Surgically treated patients underwent at least one surgical procedure involving an extraoral incision (83%); drainage of the abscess, and (89%). In severe cases cervicotomy was necessary (13%). The main role of surgery is to eliminate the source of the infection, when it can be identified, and to limit the local inflammation process with plentiful drainage of purulent accumulations and a debridement of necrotic tissue. In the current literature, the third molar is frequently the cause of complicated odontogenic infections, however in some cases the infection can persist even after tooth extraction, or be related to prosthetic surgical procedures, or implant procedures [26]. Drainage of either the neck or mediastinum is necessary, with copious lavages and placement of drains to avoid further formation of collection of purulent material. Tracheostomy was performed in three patients (2.5%). Descending necrotizing mediastinitis (DNM), is an affection due to a spread of an infective process from an origin focus located in the oropharynx that, in most of the cases, is caused by an odontogenic origin. DNM is classified as the second cause of acute mediastinitis, with a 31.82% death rate [21]. In the sample, two out of five patients with mediastinitis died in the intensive care unit (40%). Many studies have focused on the surgical treatment time in patients with mediastinitis because this seems to be correlated with the patients’ prognosis. According to Jablonski et al. early treatment reduces the risk of death in these patients [21]. Further studies are needed to confirm this data.

Although only 2.2% of total subjects required hospitalization, in 67.5% surgical treatment was necessary, 4% had severe complications, and 3.3% died. However, the mortality rate linked to oral cavity infections remains high [27,28]. It is necessary to identify risk factors and establish protocols for the management and identification of high-risk patients to reduce the rate of serious complications up to death [26,27]. Therefore, odontogenic infections today represent the second cause of acute mediastinitis [21] with a death rate of 40%. Many accesses to the emergency unit could be prevented with guidelines with risk criteria to divide patients according to the severity of the infection and with dentists trained in the management of oral cavity infections.

The limit of this retrospective study is related to the risk of underestimating or not including patients whose cause of infection was unknown at the time of their admission to the ER and was not updated on the system at the time of their medical discharge.

## 5. Conclusions

Over a 5-year period, 151 patients have been hospitalized for an odontogenic infection at Policlinico Umberto I, “Sapienza” University of Rome, and 3.97% of them have shown severe consequences such as DNI, Ludwig’s angina, or mediastinitis. A total of 116 patients received surgical treatment, including surgical drainage, securing airways, and removal of the primary focus of infection. Incidence depended only partially on socio-economic factors, loss of dental prevention, and patient risk factors. In agreement with Anderson et al. [6], access of the population to dental treatment in primary health care should be simpler, in order to reduce the burden of care on NHS hospitals for preventable dental diseases and to minimize the risk of critical complications.

## Figures and Tables

**Figure 1 ijerph-20-03469-f001:**
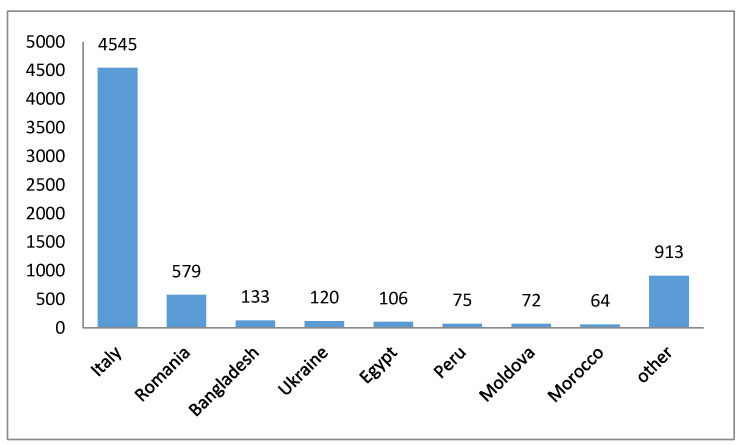
Reported nationality of the patient.

**Figure 2 ijerph-20-03469-f002:**
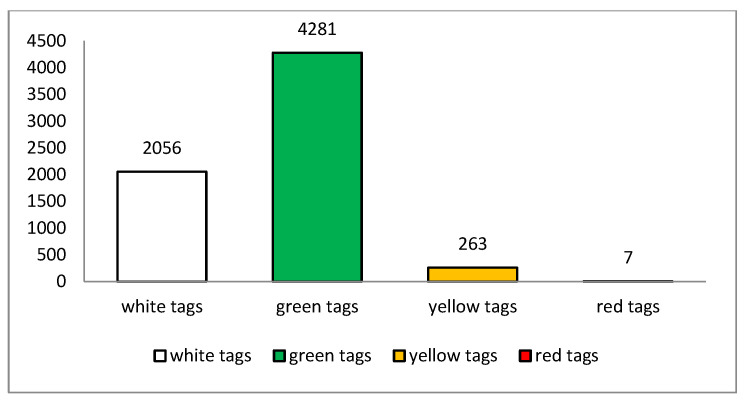
Reported the distributions of the triage tags.

**Table 1 ijerph-20-03469-t001:** Sample characteristics. Descriptive statistics performed by SPSS Software, version 26.0.

Subject Demographics					
	Odontogenic Infection Diagnosis	Hospitalized Patients	*p-*Value	Patients Surgically Treated	*p-*Value
**Sample size, n (%)**					
	6607 (10.38%)	151 (2.28%)		116 (76.82%)	
**Gender, n (%)**					
Men Women	3597 (54.4%) 3010 (45.6%)	96 (63.57%) 55 (36.42%)	*0.0254*	73 (63%) 43 (37%)	*0.0736*
**Age (yr) mean, SD ^1^**					
	40.6 ± 15.5	39.9 ± 15.8	*0.58 **	37.4 ± 16.8	*0.02 **

^1^ SD: standard deviation; *p*-value calculated by Fisher’s Exact test, * calculated with *t*-test.

**Table 2 ijerph-20-03469-t002:** Hospitalized patients: diagnosis and treatment.

	Hospitalized Patients	Patients Surgically Treated
**Sample size, n (%)**	151	116
**Diagnosis**		
Submandibular abscess DNI ^1^ Ludwig’s angina Mediastinitis	81 (53.7%) 40 (26.5%) 25 (16.5%) 5 (3.3%)	
**Surgical treatment**		
Dental extraction Extraoral drainage Cervicotomy Mediastinum drainage		103 (89%) 96 (83%) 15 (13%) 5 (4.3%)

^1^ DNI: deep neck infection.

## Data Availability

Not applicable.

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
