# Peer review of "Odontogenic-Related Head and Neck Infections: From Abscess to Mediastinitis: Our Experience, Limits, and Perspectives—A 5-Year Survey"

_ijerph, 2023, doi:10.3390/ijerph20043469_

Round 1

Reviewer 1 Report

The manuscript refers to a epidemiological retrospective observacional study concerning "Odontogenic-related head and neck infections: from abscess to mediastinitis". Overall authors conducted a study of IJERPH's readers interest thru a well conducted work. Specifically:

1. Keywords presented by authors are pertinent and avoid reader's search inconsistencies.
2. Authors obeyed the study's design requirements: Declaration of Helsinki (Line 101)and provided Ethics Approval (Lines 102-103)
3. Sample size described thru lines 130-135 is reliable and reinforces study's conclusions confidence
4. Authors described study's limits thru lines 120-123.
5. Discussion Section was well conducted, discussing the findings of the submitted manuscript with current scientific literature.

6. Authors were able to present their reality concerning the proposed subject with a reliable sample size which adds value for dissemination in the scientific community. 

Author Response

We would like to thank the reviewers for carefully reviewing our manuscript. We really appreciate the reviewer's observations.

Reviewer 2 Report

The study presented in this abstract is an epidemiological retrospective observational study that aimed to investigate the patterns, management, and surgical procedures used to treat head and neck infections caused by odontogenic infections. The study was conducted at a single center over a period of 5 years and found that out of 376940 patients who entered the emergency room, 6607 were diagnosed with odontogenic abscess (10.38%). Of these patients, 151 were hospitalized, 116 of them underwent surgical treatment (76.8%), and 6 (3.9%) developed critical conditions such as sepsis and mediastinitis.The study provides valuable insights into the prevalence and management of odontogenic infections in an emergency setting. However, it is important to note that the study is limited by its retrospective and observational nature, which may introduce bias and limit the generalizability of the findings. Additionally, the study only provides data from a single center, and it would be beneficial to see if the findings are consistent across other emergency departments.Overall, the study highlights the importance of prompt and effective treatment for odontogenic infections, as untreated or non-responsive infections can lead to severe conditions that require emergency procedures. The study's results may serve as a reminder for healthcare professionals to be vigilant when treating patients with dental affections and to be prepared to act quickly in the event of an emergency.

However, the study has several limitations. The retrospective observational design of the study may introduce bias and limit the generalizability of the findings. Additionally, the study only provides data from a single center, and it would be beneficial to see if the findings are consistent across other emergency departments. Furthermore, the study only includes patients who were hospitalized, which may lead to an underestimation of patients who were treated with oral antibiotic therapy or sent to a medical or dental clinic for treatment. Additionally, the study excluded pediatric patients, which may limit the generalizability of the findings to this population.Another limitation is that the study is based on data collection from a hospital electronic system, so there is a risk of underestimating or not including patients whose cause of infection was unknown at the admission in the ER and has not been updated on the system at the time of the medical discharge. The study also does not provide information about the follow-up of patients after being discharged from the hospital.Finally, the study only used chi-squared test to analyze the data, which may not be the most appropriate statistical analysis method for this type of study.

Author Response

Thank you for reviewing our paper. We really appreciate your comments. We have followed your advice. The limit of this investigation is to be a retrospective study and, for this reason, unfortunately the number of patients included cannot be increased. Thus, it is true that the limited size of the sample makes the results obtained less reliable. The limit of the study was described in Line 278-281. However, in the L. 200-204 and L. 207-209 the results obtained in the study were compared with the ones obtained from other single center studies in Europe. In the investigation we described 5871 patients that have been discharged from the hospital and sent home. For 517 of 6607 patients in the study (7.8%), the hospitalization was not necessary, but they underwent dental evaluation, antibiotics treatment and they referred to the dental clinic for a definitive conservative dental treatment (DCDT). We agree that these data are incomplete and that it would be useful to add details on the type of medical treatment, the type of dental treatment or the follow-up of the patients. However, being a retrospective observational study, it is not possible to extrapolate those data. This will be an inspiration for future research. Finally, we revised the statistics and added the Fisher test and T-test for the values. 

Reviewer 3 Report

Dear authors,

Thank you for submitting the manuscript entitled “Odontogenic-related head and neck infections: from abscess to mediastinitis. Our experience, limits, and perspectives. A 5-year survey”. I have carefully read it and here is my feedback:

-Introduction section, First line, you said “head and neck are often linked to an odontogenic origin”… please inform how often is this, and I assume it is a high percentage of the population.

Line treated with interdisciplinary care, but you only have one example: intensive care, please extend the information with what other types of care.

You claimed that have significant morbidity and potential mortality, so please describe how significant in percentage of cases, etc.

Again, in line 53-54 you said that it is evident that odontogenic infection represents a frequent cause of access to the emergency room but you do not give data at all, provide more information in numbers, you need to support your statements.

Remove the “by the way”, since it does not sound professional.

Lines 78 and 79, you mentioned the article accounting for 9.2% of population in Germany, please mention the country of that study and search for more articles with similar percentage so it can back-up your statement about being a significant problem.

Your manuscript needs a major revision of the English language by a native speaker.

Lines 105 to 109, there is no need to put all the diagnosis in capital letters.

Place the tables immediately after the paragraph you mention them, there is no reason to mention the table and 2 or 3 paragraphs later you include the table.

You need to standardize the tables, make them with the same style, in some you put gray background and in another you put blue background… it is common sense to put everything in the same style.

The discussion section is better than the introduction.

Please write the limitations of your investigation at the end of the discussion.

Conclusion should be shorter, no need to write 15 lines with a lot of numbers… make it shorter and concise.

Please double check the style/format of the journal and try to follow it.

Thanks.

Author Response

We would like to thank the reviewers for carefully reviewing our manuscript. We have revised the manuscript according to the suggestions of the reviewers. Revisions are shown in the revised manuscript, highlighted in yellow.

  1. Introduction section, First line, you said “head and neck are often linked to an odontogenic origin”… please inform how often is this, and I assume it is a high percentage of the population.

Epidemiological data have been added in Line 38,39,40

  1. Line treated with interdisciplinary care, but you only have one example: intensive care, please extend the information with what other types of care.

Added in Line 42

  1. You claimed that have significant morbidity and potential mortality, so please describe how significant in percentage of cases, etc.

Revised, you can find these data in Line 71-80.

  1. Again, in line 53-54 you said that it is evident that odontogenic infection represents a frequent cause of access to the emergency room but you do not give data at all, provide more information in numbers, you need to support your statements.

We added more details in Line 81-86.

  1. Remove the “by the way”, since it does not sound professional.

Done

  1. Lines 78 and 79, you mentioned the article accounting for 9.2% of population in Germany, please mention the country of that study and search for more articles with similar percentage so it can back-up your statement about being a significant problem.

We added more details in line 81-86

  1. Your manuscript needs a major revision of the English language by a native speaker.

We have provided the language revision by a native speaker

  1. Lines 105 to 109, there is no need to put all the diagnosis in capital letters.

Done

  1. Place the tables immediately after the paragraph you mention them, there is no reason to mention the table and 2 or 3 paragraphs later you include the table.

Done, Line 151,166,179

  1. You need to standardize the tables, make them with the same style, in some you put gray background and in another you put blue background… it is common sense to put everything in the same style.

We have revised the tables according to your suggestions

  1. The discussion section is better than the introduction.

Please write the limitations of your investigation at the end of the discussion.

Done

  1. Conclusion should be shorter, no need to write 15 lines with a lot of numbers… make it shorter and concise.

We agree with you, and we have revised the conclusion.

  1. Please double check the style/format of the journal and try to follow it.

Done

Thanks.

Thank you for reviewing our paper. We really appreciated your comments and suggestions.

Round 2

Reviewer 2 Report

Your article seems good but it could be improved by use graphs instead of tables.

There is no need for repetition in the result, e.g. Policlinico Umberto I, and Declaration of Helsinki... 

Author Response

We would like to thank the reviewer for the comments and the suggestions. We have provided the recommended changes and here is our response. Revisions in the manuscript are shown via track changes.

1)Your article seems good but it could be improved by use graphs instead of tables.

We have now replaced the table 2 by using graphs.

2)  There is no need for repetition in the result, e.g. Policlinico Umberto I, and Declaration of Helsinki..

Thanks for the suggestion, we have now removed those repetitions.

Kind regards
